# Vaginal Microbial Network Analysis Reveals Novel Taxa Relationships among Adolescent and Young Women with Incident Sexually Transmitted Infection Compared with Those Remaining Persistently Negative over a 30-Month Period

**DOI:** 10.3390/microorganisms11082035

**Published:** 2023-08-08

**Authors:** Supriya D. Mehta, Walter Agingu, Garazi Zulaika, Elizabeth Nyothach, Runa Bhaumik, Stefan J. Green, Anna Maria van Eijk, Fredrick O. Otieno, Penelope A. Phillips-Howard, John Schneider

**Affiliations:** 1Division of Epidemiology & Biostatistics, School of Public Health, University of Illinois Chicago, Chicago, IL 60612, USA; 2Division of Infectious Disease Medicine, College of Medicine, Rush University, Chicago, IL 60612, USA; 3Nyanza Reproductive Health Society, Kisumu P.O. Box 1764, Kenya; 4Department of Clinical Sciences, Liverpool School of Tropical Medicine, Liverpool L7 8XZ, UKpenelope.phillips-howard@lstmed.ac.uk (P.A.P.-H.); 5Kenya Medical Research Institute, Kisumu P.O. Box 1764, Kenya; 6Genomics and Microbiome Core Facility, Rush University, Chicago, IL 60612, USA; 7Departments of Medicine and Public Health Sciences, School of Medicine, University of Chicago, Chicago, IL 60637, USA; jschnei1@bsd.uchicago.edu

**Keywords:** adolescent girls and young women, vaginal microbiome, sexually transmitted infections, bacterial vaginosis, network analysis

## Abstract

A non-optimal vaginal microbiome (VMB) is typically diverse with a paucity of *Lactobacillus crispatus* and is often associated with bacterial vaginosis (BV) and sexually transmitted infections (STIs). Although compositional characterization of the VMB is well-characterized, especially for BV, knowledge remains limited on how different groups of bacteria relate to incident STIs, especially among adolescents. In this study, we compared the VMB (measured via 16S ribosomal RNA gene amplicon sequencing) of Kenyan secondary school girls with incident STIs (composite of chlamydia, gonorrhea, and trichomoniasis) to those who remained persistently negative for STIs and BV over 30 months of follow-up. We applied microbial network analysis to identify key taxa (i.e., those with the greatest connectedness in terms of linkages to other taxa), as measured by betweenness and eigenvector centralities, and sub-groups of clustered taxa. VMB networks of those who remained persistently negative reflected greater connectedness compared to the VMB from participants with STI. Taxa with the highest centralities were not correlated with relative abundance and differed between those with and without STI. Subject-level analyses indicated that sociodemographic (e.g., age and socioeconomic status) and behavioral (e.g., sexual activity) factors contribute to microbial network structure and may be of relevance when designing interventions to improve VMB health.

## 1. Introduction

Worldwide, bacterial vaginosis (BV) is the most common cause of vaginal discharge, affecting 23–29% of women in the general population [1]. This is of clinical and public health importance as BV increases the risk of HIV acquisition and is estimated to account for up to 15% of HIV infections in women [2]. BV is also associated with an increased risk of preterm birth and miscarriage [3,4] and increased prevalence and incidence of sexually transmitted infections (STIs) [5,6,7]. The most common curable STIs, *Chlamydia trachomatis* (CT), *Neisseria gonorrhoeae* (NG), and *Trichomonas vaginalis* (TV), disproportionately affect adolescent and young women [8], also contributing to increased risk of HIV acquisition and transmission and adverse birth outcomes. Given these sequelae and their co-occurrence, BV and STI prevention are a research and public health priority.

BV represents a shift away from a *Lactobacillus-dominated* vaginal microbiome (VMB) to one that is diverse (i.e., comprised of many different bacteria) and considered non-optimal, due to its association with adverse outcomes [9]. This non-optimal composition, as a molecularly defined community state type (CST), is often referred to as CST-IV. While up to 50% of women with non-optimal VMB do not have a clinical diagnosis of BV [9,10], CST-IV is associated with mucosal inflammation and an increased risk of HIV acquisition [9,11]. Less is known about how the VMB relates to incident STI in the absence of BV. Assessing this could help identify novel pathobionts or avenues for live biotherapeutics.

We sought to characterize the vaginal microbiome among secondary schoolgirls with incident STIs as compared to that of girls who remained persistently negative. Comparative studies of VMB composition often use approaches that identify differences in the relative or absolute abundance of individual taxa. Given that STI susceptibility may arise from mixed vaginal microbial interactions, we applied microbial co-occurrence network analysis to gain insights into vaginal bacteria that may interact with each other in the context of different STI and BV states. Identifying more central species (“hubs”) and the taxa they connect with could lead to targeted hypothesis generation for understanding microbiome assembly, association with pathologic states, and potential intervention targets. Additionally, network analysis based on microbial composition at the participant level can generate inferences about sociodemographic or behavioral practices that influence microbial community attributes, thus providing insights for interventions.

## 2. Materials and Methods

This study was approved by the institutional review boards of the Kenya Medical Research Institutes Scientific Ethics Review Unit (KEMRI, SERU #3215), Liverpool School of Tropical Medicine (LSTM, #15-005), and University of Illinois at Chicago (UIC, #2017-1301). Written informed parental consent and written informed assent from minors were obtained for all participants.

### 2.1. Study Design, Participants, and Sample Size

This study uses data from the Cups and Community Health (CaCHe, pronounced “Cash-Ay”) study, a prospective cohort study of adolescent girls and young women, which started when they were attending secondary school in Siaya County, western Kenya. Eligibility criteria and details of the study design have been previously reported [12]. Briefly, eligibility in the CaCHe study included attendance at a selected school, being a resident of the study area, provision of assent and parental/guardian consent, and report of established menses (having occurred at least 3 times). Participants who reported being pregnant at baseline screening were excluded. As previously reported in detail, the a priori-determined sample size was calculated to detect a 25% relative difference in cumulative prevalence of BV occurring over 30 months of follow-up, in a design of 6 repeated measurements [12]. After baseline assessment (May through June 2018), planned study visits took place at 6, 12, 18, and 30 months. The 24-month study visit was scheduled to take place in May 2020 and was missed due to the COVID-19 pandemic.

### 2.2. Detection of Bacterial vaginosis and Sexually Transmitted Infections

Participants were asked to take self-collected vaginal swabs at baseline and at each follow-up visit. At baseline, 12-, and 30-month study visits, four vaginal swabs were collected for the assessment of VMB, BV, and STIs, while at the 6- and 18-month visits, two vaginal swabs were collected for the assessment of VMB and BV. The first swab obtained was for 16S rRNA gene amplicon sequencing (VMB) using the OMNIgene Vaginal kits (OMR-130; DNA Genotek^TM^). Air-dried smears from the second swab were Gram-stained and assessed for BV according to Nugent’s criteria, with a score of 7–10 defined as BV [13]. A third vaginal swab was used for the detection of *Chlamydia trachomatis* (CT) and *Neisseria gonorrhoeae* (NG) using the GeneXpert (Cepheid, Sunnydale, CA, USA). A fourth swab was used for the detection of *Trichomonas vaginalis* (TV) using the OSOM TV antigen detection assay (Sekisui, Lexington, MA, USA). STIs (CT, NG, and TV) were treated following Kenyan National guidelines [14], and BV was treated with 2 g of tinidazole once daily for two days [15,16,17]. Treatment was documented for >95% of infections detected at each study visit [18].

### 2.3. Data Collection

Sociodemographic data and behavioral practices were collected via a self-completed tablet-based survey in the participant’s language of choice (English or DhoLuo), with assistance from study staff if needed. Socioeconomic status (SES) was assessed using abridged questions from the KEMRI health and demographic surveillance system (HDSS) household survey [19] and dichotomized as lower quintiles (1–2) and higher quintiles (3–5). At the school level, water, sanitation, and hygiene (WASH) scores ranged from 0 to 3, with 3 being the highest; a score of 3 reflected available water for handwashing, soap, and an acceptable ratio of girls to acceptable latrines (i.e., those considered in adequate condition for use) [20], which was dichotomized into 0–1 and 2–3 for the analysis. In addition to being asked about sexual activity, participants were also asked if they were forced or threatened to have sex (referred to as coerced sex) and whether they engaged in transactional sex (defined as having sex in exchange for money, items, or favors).

### 2.4. Characterization of Vaginal Microbiome

DNA extraction, library preparation, and sequencing were performed by the Genome Research Core (GRC) at the University of Illinois Chicago. DNA extraction and PCR-based library preparation of bacterial 16S rRNA gene amplicons were performed as described previously [12]. Briefly, libraries were prepared using a two-stage PCR protocol targeting the V3–V4 variable regions of bacterial 16S rRNA genes [21] and sequenced on an Illumina MiSeq instrument using a V3 kit (600 cycle chemistry). Forward and reverse reads were merged using the software package PEAR [22]. The quality and primer-trimmed sequence data were then processed using a standard bioinformatics pipeline for chimera removal, annotation, and CST typing; this processing was conducted by the University of Maryland Institute for Genomic Science [23]. Subsequently, a biological observation matrix (BIOM) was generated at the lowest taxonomic level identifiable. Vaginal community state types (CSTs) were identified in a reference dataset using the nearest centroid classification (*VA*gina*L* community state typ*E N*earest Centro*I*d cl*A*ssifier; VALENCIA) [24]. Putative contaminants were identified and removed following the application of *decontam* program in R (version 4.1.3) [25]. Raw sequence data (FASTQ files) were deposited in the National Center for Biotechnology Information (NCBI) Sequence Read Archive (SRA), under BioProject identifier PRJNA746243.

### 2.5. Construction of Analytic Data Set

Incident STI was defined as a positive test result for CT, NG, or TV, occurring at the 12- or 30-month visit, preceded by negative STI results. To minimize potential confounding from prior infection and antibiotic treatment, we used only the first incident STI and then stratified the incident STI into three categories: those occurring in the absence of BV, preceded by BV, or simultaneously occurring with BV. If a participant tested positive for an STI at 12 or at 30 months, the analysis utilized VMB data observed at the 12- or 30-month visit, respectively, rather than at prior visits given the long period of time between visits. However, we used all study visits (baseline, 6, 12, 18, and 30 months) to determine whether participants were persistently negative for BV and STI, or whether STI was preceded by BV (Appendix A). After identifying observations for the analytic sample (n = 180 persistently negative, n = 41 incident STI in the absence of BV, n = 20 incident STIs preceded by BV, and n = 14 incident STI with co-incident BV), there were 3 observations with <5000 total sequence reads which were not included in analyses (2 observations with incident STI not preceded by or simultaneous with BV and 1 observation from a participant persistently negative for BV and STI). Data points from those testing negative for BV and STI throughout observation were selected with simple random sampling stratified by time point to match the time point of incident STI (Appendix A). Since this analysis sought to gain insights into individuals with incident STIs, we did not conduct analyses on individuals with prevalent or incident BV in the absence of incident STIs, and our analyses of prevalent BV and/or STIs have been reported [12].

### 2.6. Statistical Analysis

The analysis took place in two steps: (1) microbial co-occurrence network analysis and (2) inferential analysis of participant-level microbial network characteristics in relation to participant-level demographics and behaviors, detailed below.

*(1) Microbial Co-Occurrence Network Analysis*: We conducted undirected network analysis of taxa to identify nodes and connections that differ between the outcome states: (1) 179 individuals who were persistently negative for BV and STI over 30 months of observation; (2) 39 individuals with incident STI in the absence of BV (i.e., there was no observation of BV preceding or coincident with STI; referred to as “STI with no BV”); (3) 20 individuals with incident STI that was preceded by BV (referred to as “BV before STI”). It has been recommended that a minimal sample of 20–25 observations is used in microbial network co-occurrence analysis [26], and networks are not constructed or analyzed for the 14 individuals who experienced incident STI and incident BV at the same time. Their demographic, behavioral, baseline CST, and CST at the time of infection are presented in Appendix A.

Undirected microbial co-occurrence networks were constructed separately for the three outcome states using the *NetComi* package in R [27], implementing SPRING (semi-parametric rank-based approach for inference in the graphical model) for the association measure. SPRING was selected for its advantages in estimating sparse microbial association networks, robustness to misspecification of total cell count estimate, and reliability of network metrics [28]. Prior to relative abundance estimation, data were filtered to retain taxa that contributed at least 0.01% of the total sequence reads and were present in at least 5% of observations in the analysis, resulting in the selection of 54 taxa. Within SPRING, a modified center log ratio (*mclr*) transformation was applied to address zero counts and compositionality. The number of lambda values was set to 100, with 100 repetitions. Using the *netAnalyze* function of *NetCoMi,* we report normalized centralities. Clusters of taxa within the network were generated using greedy modularity optimization (*cluster_fast_greedy* in igraph) [29], which optimizes the modularity score [30].

*(2) Network Analysis at the Participant Level*: We constructed a dissimilarity-based network, in which nodes were participants instead of taxa. Following the Bayesian multiplicative replacement of zeros and CLR transformation, Aitchison’s distance was used for the dissimilarity measure. Similarities are used as edge weights, and thus, participants with more comparable microbial network structures are arranged in closer proximity on the network graph. Properties of clustering, eigenvector centrality (the degree to which a node is connected to other highly connected nodes), and betweenness centrality (how often a node lies on the shortest path between two other nodes) were extracted. Eigenvector and betweenness centrality were selected because of their relevance to identify potentially “influential” and “gatekeeper” taxa, respectively. Network properties were compared based on participant characteristics and factors that were applicable at baseline, including intervention assignment and WASH score; and at baseline and follow-up: age, SES, sexual activity, coerced sex, transactional sex, having a boyfriend, vaginal microbiome CST, and STI etiology. Differences in distributions were assessed using the Chi-square test (with Fisher’s exact test when cell size n < 5) for categorical variables and Wilcoxon rank sum tests and Kruskal–Wallis tests for continuous variables.

*Sensitivity Analyses.* Because the majority of incident STIs were *C. trachomatis*, we attempted to construct a subject-level network for the 21 participants with incident CT in the absence of BV, TV, and NG. No optimum number of clusters could be defined; therefore, sparsification of the network to extract subject-level components was not attempted due to the potential unreliability of findings.

## 3. Results

### 3.1. Characteristics of Study Sample and Microbiome Composition

Sociodemographic and behavioral practices at baseline did not differ between participants who remained negative for BV and STI throughout the follow-up compared to those with incident STI (Table 1). However, characteristics at the time of the incident STI varied considerably with greater frequency of being sexually active, having a boyfriend, and having ever been pregnant.

The most common taxa with the highest mean relative abundance were *L. crispatus* and *L iners*, correlating with CST-I and CST-III, respectively (Figure 1). Overall, the VMB composition was significantly different between participants who remained persistently negative for BV and STIs as compared to those with incident STI, with or without BV (Figure 2A). While the majority of persistently negative participants had VMB of CST-I (*L. crispatus* dominated), *L. crispatus* was substantially depleted from the VMB of participants with STI, even in the absence of BV, but was uncommon and at very low relative abundance in the VMB of those with BV prior to or simultaneously occurring with incident STI (Figure 2B,C). Individual taxa presence and relative abundance distribution by BV and STI status are given in Appendix A.

### 3.2. Results of Microbial Co-Occurrence Network Analysis: Differences in Network Properties and Centralities for Participants with Incident STI Compared to Persistently Negative Participants

In the VMB of 179 participants who were persistently negative for STIs and BV, 3 network component clusters were identified with 1 having 24 taxa, 1 having 2 taxa, and 5 singleton taxa (Table 2). There were 4 network component clusters identified among 39 participants with incident STI and no BV: 2 components with 9 taxa each, 1 with 4 taxa, 1 with 3 taxa, and 6 singleton taxa. In keeping with this, other network metrics (relative largest connected component size, clustering coefficient, positive edge percentage, and average path lengths) also reflected a more connected vaginal microbial network for participants who remained persistently negative for STI and BV as compared to participants with incident STI and no BV. The network properties and differences are stark in the network plots (Figure 3), where the varying taxa of central importance, clusters, and connections are highlighted. Some similar trends were seen, with lesser clustering coefficient and positive edge percentage, for those with BV prior to incident STI in comparison to participants who remained negative for BV and STI throughout follow-up (Appendix A).

*Centralities* (Table 3). In the microbial co-occurrence network of individuals who remained persistently negative for STIs and BV, the taxa with the highest eigenvector centralities were as follows (in descending order): *Prevotella melaninogenica* (present in 4.0%, with a mean relative abundance (RA) of 6.68% among samples where present), *Gemella haemolysans/Gemella asaccharolytica* (present in 10.2%, with a mean RA of 2.61% where present), *Fannyhessea vaginae* (*Atopobium*; present in 13.6%, with a mean RA of 4.02% where present) bacterial-vaginosis-associated bacterium 1 (BVAB1, present in 4.5%, with a mean RA of 6.02% where present), and *Sneathia amnii* (present in 4.0%, with a mean RA of 3.86% where present). These were the same 5 taxa with the highest closeness centrality and betweenness centrality, though with *Fusobacterium nucleatum* (present in 10.2%, with a mean RA of 2.54% where present) rather than BVAB1 for highest betweenness centrality. Notably, these taxa with the highest centralities were not mirrored in the vaginal microbial co-occurrence network of participants with incident STI in the absence of BV. The only taxon with high centrality found in both sub-groups was BVAB1. Taxa with the highest centralities for those with incident STI and no BV were *Staphylococcus hominis* (present in 23.1%, with a mean RA of 0.73% where present), *Fusobacterium equinum* (present in 7.7%, with a mean RA of 2.00% where present), *Lactobacillus jensenii* (present in 17.9%, with a mean RA of 1.94% where present), and *Veillonella* (present in 20.5%, with a mean RA of 5.64% where present). According to the Jaccard index, the degree (*p* = 0.017), eigenvector (*p* = 0.026), and closeness (*p* = 0.017) centralities exhibited statistically significant differences between the two groups.

In the VMB of 20 participants with BV prior to STI, taxa with the highest centrality values were again strikingly different from participants who remained persistently negative throughout follow-up, though no centrality differences reached statistical significance by Jaccard index *p*-value, possibly due to the small sample size of those with BV prior to incident STI (Appendix A). Taxa with the highest centrality among participants with BV prior to incident STI differed from those with incident STI in the absence of BV, and the taxa with the highest values differed across centrality measures.

### 3.3. Results of Participant-Level Network Analysis: Network Properties Differ by Sociodemographic, Behavioral, and VMB Composition

In subject-level network analysis, nodes are the individual participants, but they are connected based on VMB composition. Therefore, it is unsurprising that subject-level network properties (i.e., centralities and component clusters) vary according to VMB composition (Table 4). However, subject-level network properties also varied by sociodemographic and behavioral factors. Among 179 individuals persistently negative for STIs and BV, betweenness centrality was increased for those with baseline values of higher SES, having experienced coerced sex, and having a boyfriend, while eigenvector centrality was increased among participants assigned to menstrual cup intervention, and those with younger age at follow-up. Eigenvector centrality was greatest among participants with vaginal CST-I (*L. crispatus* dominated) at baseline and at follow-up, while betweenness centrality was increased among those with vaginal CST-I and CST-III (*L. iners* dominated) at follow-up. For 39 participants with incident STI and no BV, betweenness centrality did not vary by any factors examined, while eigenvector centrality was highest for those with CST-III at follow-up and those with *C. trachomatis* etiology.

Network component clusters (Figure 4A, Table 5) for participants who were persistently STI- and BV-negative varied by vaginal CST at baseline and follow-up, with the majority (92.3%) of network cluster 2 observations having CST-I. Network cluster 3 was also predominantly CST-I (78.5%), while network cluster 1 was majority CST-III (69.4%) and CST-IV (27.8%). Network cluster 2 participants were also more likely to be younger age at follow-up, but no other sociodemographic or behavioral factors varied by network cluster for persistently STI- and BV-negative participants. Among participants with incident STI without BV (Figure 4B, Table 5), network clusters varied by vaginal CST at follow-up but not at baseline, with cluster 1 being predominantly (87.5%) CST-IV, cluster 2 being majority CST-III (60.9%) and CST-I (30.4%), and cluster 3 being 87.5% CST-III. Network clusters of those with incident STI without BV did not differ by age, SES, or behavioral factors at follow-up, but varied by baseline WASH score, sexual activity, experience of coerced sex, and report of transactional sex. Sexual activity, coerced sex, and transactional sex were more commonly reported by participants in network clusters 1 and 3 and may represent differential exposure to penile microbiomes. Participants in these clusters were also more likely to originate from school areas with lower WASH scores, another indicator of community-level SES.

## 4. Discussion

We identified differing key taxa in the VMB networks of adolescent and young women with incident STIs as compared to those who remained negative for BV and STI over a 30-month period of observation. Overall, the VMB showed decreased connectivity in individuals with incident STIs compared to those who remained persistently negative, and the taxa that were most central differed between those with incident STIs compared to those who remained negative. Secondly, participant-level network structure based on VMB composition varied by sociodemographic and behavioral factors, and network clusters correlated with molecular CSTs but were not redundant with them.

The differences in the VMB network structures and properties have implications for bacterial community function. In general, the VMB of those with incident STIs showed decreased connectivity, as reflected in lower measures of clustering, smaller largest connected components, and reduced edge positivity percentage. A microbial network with lower connectivity may reflect less “collaboration” or more competition among the taxa [31] and has been associated with pathogenic states [32,33]. The taxa identified as having central importance varied by infection status. Notably, these taxa were not those that were driving overall compositional differences (Figure 2) or with the highest presence and relative abundance (Appendix A). These key taxa, even low-abundance genera, may have central roles, possibly related to gatekeeping or communication [34]. The highest centrality taxa in the VMB of persistently negative participants have been consistently associated with BV [9]: *Fannyhessea vaginae* (*Atopobium*), *Prevotella melaninogenica, Gemella,* BVAB1, and *Sneathia amnii.* In the context of a disease-free state and majority with optimal VMB (58% CST-I at baseline and 50% CST-I at follow-up), the identification of these taxa as key players may be revealing their latent pathobiont nature. Conversely, the taxa with the highest centralities in the VMB of participants with incident STI identified *L. jensenii, Staphylococcus hominis*, *Fusobacterium equinum*, *Veillonella*, and BVAB1. Like BVAB1, *Veillonella* species have also been identified in conjunction with BV, having a potential role in weakening the cervicovaginal epithelial barrier [35] and has been associated with BV treatment failure [36]. *L. jensenii* is the dominant taxon in CST-V, an uncommon CST in our dataset, as in others, and is generally considered beneficial in the vaginal microbiome [34]. There is evidence for a protective role of *L. jensenii*. For example, in a study of 220 women of varying race/ethnicity in the United States, Srinivasan et al. observed an inverse relationship between BV and *L. jensenii* [37], and in laboratory studies, *L. jensenii* has been shown to inhibit gonococcal adherence to epithelial cells [38]. In the context of our study, the greater centrality of *L. jensenii* may represent a change in composition or perturbation of *L. jensenii* homeostasis. To our knowledge, *Staphylococcus hominis* and *Fusobacterium equinum* have not been previously associated with BV, STIs, or other VMB-related conditions. In this analysis, they may represent opportunistic colonization in the setting of STI. Longitudinal network studies that incorporate bacterial function along with composition and clinical outcomes will be necessary to disentangle whether and how the centrality of taxa changes as a function of infection.

There was a lower proportion of positive edge percentages in the VMB of participants with incident STIs, especially within the largest connected component. The higher prevalence of positive edges in the VMB of persistently negative participants suggests potentially greater sharing of environmental spaces or conditions or greater sharing of bacterial products [39]. As described by Baquero et al., this collaboration could improve “homeostatic power”, enabling the established community to be more resilient to “foreign organisms” [40]. The larger proportion of negative edges in the VMB of those with STI may indicate greater competition among the bacteria. The differences in network metrics in the VMB of those with STI may also suggest a potential underlying environmental imbalance. VMB community perturbation likely occurs prior to STI acquisition *and* as a result of STI acquisition. We analyzed the network structure at the time of acquisition, given the time interval between STI testing and microbiome assessments. Prospective microbiome–STI studies with frequent sampling would be able to shed more light on the temporal associations.

We are unaware of other vaginal microbial co-occurrence networks related to incident STIs with which we can compare our results. Antibiotic treatment of *C. trachomatis* has been shown to alter the VMB composition in a potentially non-optimal way [41], and microbial network analysis could help understand this effect by characterizing hub or connection disruptions and VMB restructuring in this context. For example, this might include serial network construction based on sampling of VMB at the time of infection detection, immediately following STI treatment, and again at 4 to 8 weeks, which would allow evaluation of which hubs and articulation points are disrupted alongside antimicrobial treatment of STIs. This would also inform whether the VMB restructures differently in terms of how the taxa interact with each other, building on the knowledge of compositional changes. In conjunction with bacterial function studies, this could provide potent insights into new biotherapeutic avenues for increasing VMB resilience to STIs.

The results of our subject-level microbial network analysis reinforce the knowledge that the VMB is shaped by the environment. Subject trait-driven network characteristics (i.e., the centrality of individuals and network clusters based on microbiome composition) varied by sociodemographic factors and sexual exposures and may represent different sexual networks (i.e., the connections among individuals defined by sexual relationships). These factors could directly affect sexual partner selection (such as age, SES, and proximity) or may represent norms and beliefs around hygiene practices and sexual practices (such as condom use and multiple partners), which drive partner type and selection. These are novel analyses in that they potentially capture a proxy for sexual mixing, and future studies should integrate traditional sexual network analysis with microbiome network analysis to characterize and quantify how they overlap. As demonstrated with a simulation study, Kenyon et al. found that populations with higher heterosexual connectivity had a higher population-level prevalence of BV than did communities with lower sexual network connectivity [42]. This is rational, as the authors summarize that a preponderance of data demonstrates the sharing of the genital microbiota between individuals, the transmission of STIs along sexual networks, and therefore, the transmission of genital microbiota within a sexual network in a similar manner. Consideration and incorporation of the microbiome-sexual network in biobehavioral interventions may contribute to their effectiveness.

For our network analyses, incident STI was a composite of chlamydia, gonorrhea, and trichomoniasis. With just over half of incident STIs being CT, we conducted supplemental network analyses among these participants which revealed limited information, likely due to the small sample size. As a follow-up of our cohort is continuing through 72 months, and the incidence of STIs has increased over time, we may reach a sufficient sample size to be able to detect differences in microbial co-occurrence networks between incident STIs of different etiology. Characterizing the relationship of these pathogens to the contextual microbiota, and ideally also in relation to host immune-related processes, could identify avenues for biotherapeutics and vaccine development [43]. For *C. trachomatis*, such studies may also shed light on biological factors that influence clearance rather than persistence [44].

### Limitations

There were small sample sizes for incident STIs preceded by BV (n = 20), co-incident STIs and BV (n = 14), and for singular incident etiologies (CT alone, TV alone, and NG alone). However, the characterization of VMB is nascent in relation to incident STIs, especially among adolescents and young women. While a body of literature has established VMB compositional differences of women with STIs and/or BV compared to women without infections, our microbial network co-occurrence approach uncovered novel findings that can be examined in larger sample sizes. The number of individuals and a number of observations per individual can influence the community structure [8], and as a follow-up of the cohort continues through 72 months, we will have the opportunity to expand analysis, with future plans to examine the temporal stability of these microbial co-occurrence networks. Sexual behaviors were likely to have been underreported as we previously reported [12], due to the stigma associated with this. Disclosure of such information may result in, or be perceived to result in, potential harm. To minimize this, no identifying information was collected in conjunction with research data, and at the time of data collection, extensive efforts were made to ensure privacy and confidentiality [12]. Regarding dissemination, to minimize potential negative consequences in the community, we do not report the schools involved in the study or the home areas of the participants. As with any longitudinal study, outcomes (i.e., BV or STI) may have occurred and been resolved prior to baseline observation. However, Kenya relies on syndromic management of vaginal discharge, and a high proportion of BV and STI cases are asymptomatic and thus would not have been treated.

## 5. Conclusions

Our study identified vaginal taxa with central importance to microbial community network structure, which differ between adolescent and young women who remained persistently negative for STIs and BV in comparison to those who acquired STIs. These key taxa may have an important role in bacterial community communication, competition, homeostasis, and collective function for preventing or permitting infection. Longitudinal studies that combine bacterial function and network analyses, with sexual network and sociodemographic and behavioral information, at acquisition and treatment inflection points could contribute to advancing biotherapeutic and behavioral interventions to disrupt STI transmission.

## Figures and Tables

**Figure 1 microorganisms-11-02035-f001:**
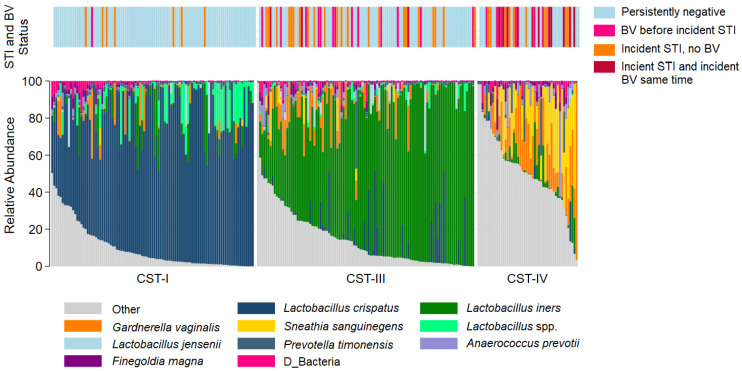
Stacked bar chart of the relative abundance of the vaginal taxa with the greatest mean relative abundance for individuals by community type. Legend: Observations from 249 samples are sorted by molecular community state type (CST). The relative abundance of the taxa with the highest mean relative abundance is shown (0–100%; y-axis), with individual subjects represented by individual bars, separated by CST (x-axis). The bar at the top represents STI and BV outcome status.

**Figure 2 microorganisms-11-02035-f002:**
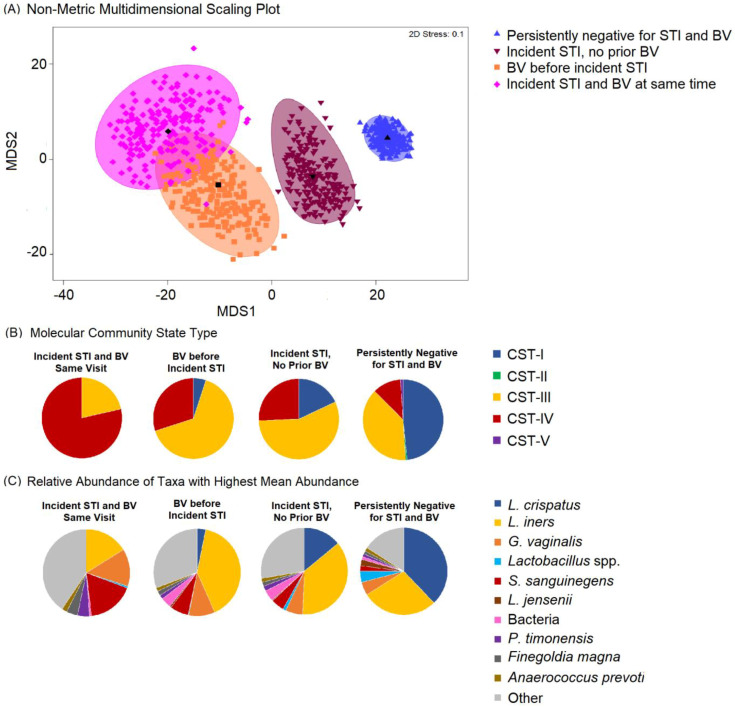
Non-metric dimensional scaling plot for each of the four outcome states for incident sexually transmitted infection (STI) or bacterial vaginosis (BV) and distribution of Community State Type (CST) and taxa relative abundance, N = 249. Legend: (**A**) The four different colors represent the four outcome states for STI and BV. STI is a composite of infection with any of *C. trachomatis*, *N. gonorrhoeae,* and *T. vaginalis.* Blue = negative for STIs and BV throughout follow-up; maroon = incident STI in the absence of BV; orange = BV positive prior to incident STI; pink = incident STI simultaneous with incident BV. Each colored mark indicates 1 of 200 bootstrappings of the dataset. The matching shaded area represents 95% coverage. The black symbol at the center of each colored shape represents the average centroid of the 200 bootstraps. (**B**) The pie charts below show the distribution of CST, aligned to outcome states. (**C**) The pie charts below show the distribution of taxa with the highest mean relative abundance, aligned to outcome states.

**Figure 3 microorganisms-11-02035-f003:**
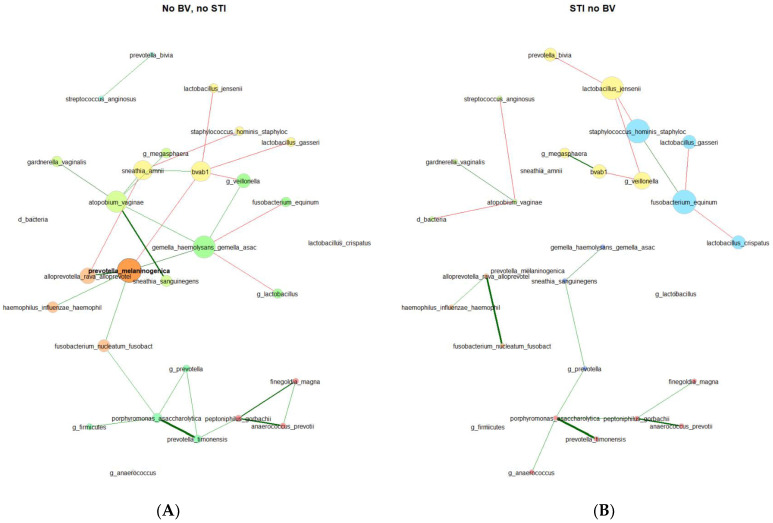
Microbial co-occurrence plots for (**A**) participants who were persistently negative for STI and BV (N = 179) and (**B**) participants with incident STI in the absence of BV (N = 39). Legend: Undirected plots are plotted on a union layout with single nodes removed for ease of visibility. Nodes represent taxa, with colors identifying connected components, with node size scaled to eigenvector centrality. Correlations are depicted with red (negative) or green (positive) connectors, with a width of the line proportional to the correlation.

**Figure 4 microorganisms-11-02035-f004:**
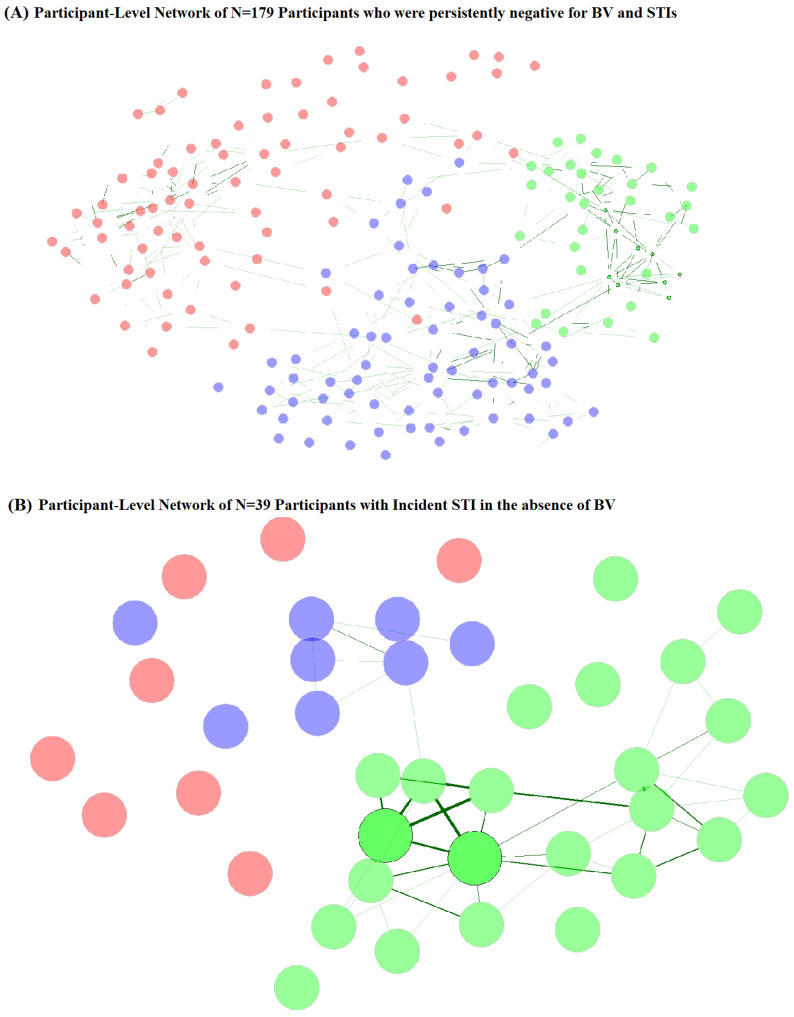
Participant-level network plots for (**A**) participants who were persistently negative for STI and BV (N = 179) and (**B**) participants with incident STI in the absence of BV (N = 39). Legend: Undirected plots are filtered for ease of visibility. Nodes represent individuals, with colors identifying clusters. Similarities are depicted with green connectors, with a width of the line proportional to the similarity.

**Table 1 microorganisms-11-02035-t001:** Distribution of baseline and time-updated characteristics of study sample by outcome status.

Characteristics ^1^	Persistently BV and STI Negative, N = 179 n (%)	Incident STI with No Prior BV, N = 39 n (%)	Incident STI with Prior BV, N = 20 n (%)	Incident STI and BV at Same Time, N = 14 n (%)
** *At Baseline* **				
Randomization status				
Control arm	92 (51.4)	16 (41.0)	13 (65.0)	11 (78.6)
Cup arm	87 (48.6)	23 (59.0)	7 (35.0)	3 (21.4)
Median age in years				
<16.9 years	101 (56.4)	21 (53.9)	11 (55.0)	4 (28.6)
≥16.9 years	78 (43.6)	18 (46.1)	9 (45.0)	10 (71.4)
Socioeconomic status score				
Highest quintiles	129 (72.1)	26 (66.7)	12 (60.0)	11 (78.6)
Lowest quintile	50 (27.9)	13 (33.3)	8 (40.0)	3 (21.4)
Water, sanitation, and hygiene score				
Higher	70 (39.1)	16 (41.0)	9 (45.0)	6 (42.9)
Lower	109 (60.9)	23 (59.0)	11 (55.0)	8 (57.1)
Sexually active				
No	132 (74.2)	24 (61.5)	13 (65.0)	10 (71.4)
Yes	46 (25.8)	15 (38.5)	7 (35.0)	4 (28.6)
Ever engaged in sex in exchange for money, favors, or things				
INo	162 (91.0)	33 (84.6)	17 (85.0)	13 (92.9)
Yes	16 (9.0)	6 (15.4)	3 (15.0)	1 (7.1)
Ever been forced, tricked, or coerced to have sex				
No	146 (82.0)	26 (66.7)	18 (90.0)	10 (71.4)
Yes	32 (18.0)	13 (33.3)	2 (10.0)	4 (28.6)
Currently has a boyfriend				
No	173 (96.7)	36 (92.3)	17 (85.0)	13 (92.9)
Yes	6 (3.3)	3 (7.7)	3 (15.0)	1 (7.1)
Community State Type (CST)				
CST-I	101 (57.7)	14 (35.9)	4 (20.0)	6 (42.9)
CST-II	4 (2.3)	1 (2.6)	1 (5.0)	0 (0.0)
CST-III	53 (30.3)	19 (48.7)	7 (35.0)	6 (42.9)
CST-IV	12 (6.9)	3 (7.7)	8 (40.0)	1 (7.1)
CST-V	5 (2.9)	2 (5.1)	0 (0.0)	1 (7.1)
** *At time of Incident STI (or time of sampling if persistently negative)* **	**Persistently BV and STI Negative, N = 179** **n (%)**	**Incident STI with no prior BV, N = 39** **n (%)**	**Incident STI with Prior BV, N = 20** **n (%)**	**Incident STI and BV at same time, N = 14** **n (%)**
Time of incident STI or time of sampling if persistently negative				
12 months	74 (41.3)	22 (56.4)	6 (30.0)	3 (21.4)
30 months	105 (58.7)	17 (43.6)	14 (70.0)	11 (78.6)
Median age in years				
<18.8 years	95 (53.1)	24 (61.5)	8 (40.0)	4 (28.6)
≥18.8 years	84 (46.9)	15 (38.5)	12 (60.0)	10 (71.4)
Socioeconomic status score				
Median or higher	102 (57.6)	26 (68.4)	9 (47.4)	6 (46.2)
Below median	75 (42.4)	12 (31.6)	10 (52.6)	7 (53.8)
Ever sexually active				
No	75 (42.6)	12 (32.4)	4 (21.1)	2 (14.3)
Yes	101 (57.4)	25 (67.6)	15 (78.9)	12 (85.7)
Ever engaged in sex in exchange for money, favors, or things				
No	161 (91.0)	31 (81.6)	17 (89.5)	11 (78.6)
Yes	16 (9.0)	7 (18.4)	2 (10.5)	3 (21.4)
Ever been forced, tricked, or coerced to have sex				
No	151 (85.8)	28 (73.7)	18 (94.7)	10 (71.4)
Yes	25 (14.2)	10 (26.3)	1 (5.3)	4 (28.6)
Currently has a boyfriend				
No	142 (80.2)	24 (63.2)	10 (52.6)	6 (42.9)
Yes	35 (18.9)	14 (36.8)	9 (47.4)	8 (57.1)
Ever been pregnant 2				
No	103 (97.2)	23 (85.2)	10 (71.4)	10 (83.3)
Yes	3 (2.8)	4 (14.8)	4 (28.6)	2 (16.7)
Community State Type				
CST-I	89 (49.7)	7 (18.0)	1 (5.0)	0 (0)
CST-II	1 (0.6)	0 (0)	0 (0)	0 (0)
CST-III	66 (36.9)	22 (56.4)	13 (65.0)	3 (21.4)
CST-IV	21 (11.7)	10 (25.6)	6 (30.0)	11 (78.6)
CST-V	2 (1.1)	0 (0)	0 (0)	0 (0)

^1^ Not all cells sum to N due to missing data. ^2^ Ever pregnant was asked only to those who reported being sexually active; hospitalization for pregnancy in the past 6 months also supplemented this, in some cases leading to a number of responses to “ever pregnant” being greater than the number reporting being sexually active.

**Table 2 microorganisms-11-02035-t002:** Distribution of network properties by outcome: persistently negative for sexually transmitted infection (STI) and bacterial vaginosis (BV) and incident STI in the absence of BV.

Network Properties	Persistently Negative, No STI and No BV N = 179	Incident STI, No Prior BV N = 39
**Components and sizes**	3 components 24 (1) 2 (1) 1 (5)	4 components 9 (2) 4 (1) 3 (1) 1 (6)
**Largest connected component (LCC)**		
Relative LCC size	0.774	0.290
Clustering coefficient	0.277	0
Modularity	0.543	0.398
Positive edge percentage	71.4	25.0
Edge density	0.101	0.222
Natural connectivity	0.054	0.159
Vertex connectivity	1	1
Edge connectivity	1	1
Average dissimilarity	0.680	0.697
Average path length	3.66	2.84
**Whole network**		
Number of components	7	10
Clustering coefficient	0.277	0
Modularity	0.563	0.719
Positive edge percentage	72.4	61.9
Edge density	0.062	0.045
Natural connectivity	0.04	0.039

**Table 3 microorganisms-11-02035-t003:** Distribution of network centralities by outcome: persistently negative for STI and BV and incident STI in the absence of BV.

	**Network Centrality Measures ^1^**		
**Taxon**	**Persistently Negative,** **No STI and No BV** **N = 179**	**Incident STI,** **no Prior BV** **N = 39**	**Absolute Difference**
	**Degree Centrality**		
*Fannyhessea vaginae (Atopobium)*	0.167	0	0.167
*Prevotella melaninogenica*	0.167	0	0.167
*Gemella haemolysans/Gemella asaccharolytica*	0.167	0	0.167
Bacterial vaginosus associated bacterium 1 (BVAB1)	0.167	0.067	0.1
*Sneathia amnii*	0.133	0	0.133
*Lactobacillus jensenii*	0.033	0.1	0.067
*Fusobacterium equinum*	0.033	0.1	0.067
*Veillonella*	0.067	0.067	0
*Staphylococus hominus*	0.033	0.067	0.034
	**Betweenness Centrality**		
*Prevotella melaninogenica*	0.593	0	0.593
*Fusobacterium nucleatum*	0.443	0	0.443
*Porphyromonas asaccharolytica*	0.423	0	0.423
*Gemella haemolysans/Gemella asaccharolytica*	0.387	0	0.387
*Fannyhessea vaginae* (*Atopobium)*	0.273	0	0.273
*Lactobacillus jensenii*	0	0.679	0.679
*Staphylococcus hominis*	0	0.536	0.536
*Fusobacterium equinum*	0	0.464	0.464
*Veillonella*	0	0.429	0.429
BVAB1	0.178	0.25	0.072
	**Closeness Centrality**		
*Prevotella melaninogenica*	0.525	0	0.525
*Gemella haemolysans/Gemella asaccharolytica*	0.505	0	0.505
*Fannyhessea vaginae* (*Atopobium)*	0.49	0	0.49
BVAB1	0.48	0.483	0.003
*Sneathia amnii*	0.458	0	0.458
*Lactobacillus jensenii*	0.32	0.614	0.294
*Fusobacterium equinum*	0.33	0.583	0.352
*Staphylococcus hominis*	0.31	0.577	0.24
*Veillonella*	0.387	0.533	0.146
	**Eigenvector Centrality**		
*Prevotella melaninogenica*	1	0	1
*Gemella haemolysans/Gemella asaccharolytica*	0.912	0	0.912
*Fannyhessea vaginae* (*Atopobium)*	0.865	0	0.865
BVAB1	0.787	0.465	0.779
*Sneathia amnii*	0.755	0	0.755
*Staphylococcus hominis*	0.221	1	0.779
*Fusobacterium equinum*	0.263	0.999	0.736
*Lactobacillus jensenii*	0.23	0.943	0.68
*Veillonella*	0.483	0.643	0.16

^1^ Network centrality measures are normalized for within-sample comparison. For each centrality measure, the top 5 taxa for each sub-group are reported and are presented first for the persistently negative sub-group and then for the sub-group of incident STI in the absence of BV. One taxon, BVAB1, is the highest-ranked centrality measure in both sub-groups. Shading is applied to the absolute difference column to facilitate the reading of taxa with greater (darker intensity shading) absolute difference vs. taxa with lower (lighter shading) differences.

**Table 4 microorganisms-11-02035-t004:** Distribution of characteristics by network centrality measures by outcome status: persistently negative for sexually transmitted infection (STI) and bacterial vaginosis (BV) and incident STI with no BV.

	Persistently STI and BV Negative, N = 179 ^1^		Incident STI with No BV, N = 39 ^2^	
	Betweenness Centrality Mean (SD)	Eigenvector Centrality Mean (SD)	Betweenness Centrality Mean (SD)	Eigenvector Centrality Mean (SD)
Mean (standard deviation)	0.022 (0.038)	0.079 (0.155)	0.055 (0.100)	
*Characteristics at Baseline*				
Intervention arm		*		
Control	0.023 (0.043)	**0.082 (0.173)**	0.086 (0.145)	0.338 (0.303)
Menstrual cup	0.022 (0.032)	**0.076 (0.135)**	0.033 (0.042)	0.200 (0.184)
Water, sanitation, and hygiene score				*
Higher score	0.024 (0.039)	0.083 (0.151)	0.047 (0.096)	**0.200 (0.254)**
Lower score	0.022 (0.038)	0.076 (0.159)	0.060 (0.105)	**0.296 (0.237)**
Median age				
<16.9 years	0.022 (0.039)	0.080 (0.159)	0.041 (0.102)	0.226 (0.262)
16.9 years or older	0.024 (0.037)	0.077 (0.152)	0.070 (0.099)	0.292 (0.228)
Socioeconomic status score	*			
Higher quartiles	**0.025 (0.039)**	0.082 (0.161)	0.072 (0.116)	0.301 (0.277)
Lowest quartile	**0.018 (0.034)**	0.072 (0.141)	0.021 (0.046)	0.158 (0.127)
Sexually active				
No	0.020 (0.038)	0.075 (0.152)	0.060 (0.104)	0.291 (0.262)
Yes	0.030 (0.038)	0.092 (0.168)	0.046 (0.097)	0.201 (0.214)
Experienced coerced sex	**			
No	**0.020 (0.037)**	0.077 (0.150)	0.055 (0.101)	0.280 (0.257)
Yes	**0.035 (0.039)**	0.089 (0.183)	0.054 (0.103)	0.209 (0.224)
Had transactional sex				
No	0.021 (0.036)	0.075 (0.146)	0.060 (0.108)	0.286 (0.255)
Yes	0.039 (0.051)	0.124 (0.246)	0.023 (0.025)	0.091 (0.065)
Has a boyfriend	**			
No	**0.021 (0.037)**	0.074 (0.144)	0.056 ().104)	0.249 (0.241)
Yes	**0.068 (0.048)**	0.208 (0.356)	0.037 (0.034)	0.341 (0.352)
Vaginal Community State Type (CST)		***		
CST-I (*L. crispatus* dominated)	0.024 (0.040)	**0.091 (0.136)**	0.067 (0.112)	0.365 (0.273)
CST-III (*L. iners* dominated)	0.024 (0.035)	**0.053 (0.159)**	0.053 (0.104)	0.230 (0.229)
CST-IV (mixed)	0.014 (0.039)	**0.015 (0.017)**	0.061 (0.092)	0.088 (0.114)
*Characteristics at Follow-Up*				
Median age		******		
Below 18.8 years	0.022 (0.039)	**0.106 (0.180)**	0.046 (0.095)	0.252 (0.248)
18.8 years or older	0.023 (0.037)	**0.049 (0.115)**	0.068 (0.110)	0.263 (0.251)
SES score		*****		
Above median	0.027 (0.044)	**0.105 (0.191)**	0.035 (0.077)	0.241 (0.219)
Below median	0.016 (0.025)	**0.040 (0.069)**	0.087 (0.135)	0.293 (0.313)
Sexually active				
No	0.019 (0.034)	0.079 (0.159)	0.019 (0.029)	0.195 (0.144)
Yes	0.025 (0.039)	0.078 (0.154)	0.068 (0.119)	0.230 (0.285)
Experienced coerced sex				
No	0.024 (0.039)	0.081 (0.160)	0.047 (0.086)	0.253 (0.235)
Yes	0.013 (0.024)	0.064 (0.124)	0.064 (0.136)	0.269 (0.298)
Had transactional sex				
No	0.022 (0.037)	0.074 (0.144)	0.054 (0.106)	0.258 (0.250)
Yes	0.030 (0.042)	0.115 (0.243)	0.041 (0.076)	0.255 (0.264)
Has a boyfriend				
No	0.022 (0.038)	0.081 (0.154)	0.045 (0.096)	0.267 (0.243)
Yes	0.025 (0.034)	0.064 (0.162)	0.063 (0.109)	0.240 (0.268)
CST at follow-up	*	*******		******
CST-I (*L. crispatus* dominated)	**0.024 (0.043)**	**0.145 (0.120)**	0.040 (0.078)	**0.189 (0.143)**
CST-III (*L. iners* dominated)	**0.025 (0.035)**	**0.017 (0.040)**	0.078 (0.122)	**0.358 (0.271)**
CST-IV (mixed)	**0.010 (0.021)**	**0.002 (0.005)**	0.014 (0.016)	**0.081 (0.085)**
Network clusters		*******		*******
1	0.021 (0.032)	**0.006 (0.014)**	0.024 (0.021)	**0.073 (0.081)**
2	0.029 (0.053)	**0.272 (0.241)**	0.070 (0.124)	**0.367 (0.264)**
3	0.021 (0.033)	**0.047 (0.049)**	0.040 (0.063)	**0.122 (0.058)**
STI etiology (comparison restricted to sole infections)	NA	NA		*
*C. trachomatis* (n = 21)			0.080 (0.125)	**0.359 (0.279)**
*N. gonorrhoeae* (n = 3)			0.021 (0.021)	**0.111 (0.017)**
*T. vaginalis* (n = 11)			0.031 (0.063)	**0.167 (0.146)**

^1^ *p*-value by Wilcoxon rank sum test for 2 categories or Kruskal–Wallis test for 3 categories (CST and STI etiology). ^2^ *p*-value by Chi-square test or Fisher’s exact test when cell size <5. * *p*-value < 0.05; ** *p*-value < 0.01; *** *p*-value < 0.001. Comparisons with *p*-value < 0.05 are bolded.

**Table 5 microorganisms-11-02035-t005:** Distribution of participant characteristics by network clusters (based on microbiome composition).

	Persistently STI and BV Negative, N = 179			Incident STI with No BV, N = 39		
	Cluster 1, N = 73 n (%)	Cluster 2, N = 39 n (%)	Cluster 3, N = 67 n (%)	Cluster1, N = 8 n (%)	Cluster 2, N = 23 n (%)	Cluster 3, N = 8 n (%)
*Baseline Characteristics*						
Intervention arm						
Control	39 (53.4)	20 (51.3)	33 (49.3)	2 (25.0)	11 (47.8)	3 (37.5)
Menstrual cup	34 (46.6)	19 (48.7)	34 (50.7)	6 (75.0)	12 (52.2)	5 (62.5)
WASH score						*****
Lower	27 (37.0)	15 (38.5)	28 (41.8)	**6 (75.0)**	**5 (21.7)**	**5 (62.5)**
Higher	46 (63.0)	24 (61.5)	39 (58.2)	**2 (25.0)**	**18 (78.3)**	**3 (37.5)**
Median age						
<16.9 years	37 (50.7)	23 (59.0)	41 (61.2)	5 (62.5)	12 (52.2)	4 (50.0)
16.9 years and older	36 (49.3)	16 (41.0)	26 (38.8)	3 (37.5)	11 (47.8)	4 (50.0)
Socioeconomic score						
Lower	55 (75.3)	26 (66.7)	48 (71.6)	6 (75.0)	17 (73.9)	3 (37.5)
Higher	18 (24.7)	13 (33.3)	19 (28.4)	2 (25.0)	6 (26.1)	5 (62.5)
Sexually active						*****
No	50 (68.5)	28 (71.8)	54 (81.8)	**2 (25.0)**	**18 (78.3)**	**4 (50.0)**
Yes	23 (31.5)	11 (28.2)	12 (18.2)	**6 (75.0)**	**5 (21.7)**	**4 (50.0)**
Experienced coerced sex						*****
No	55 (75.3)	33 (84.6)	58 (87.9)	**4 (50.0)**	**19 (82.6)**	**5 (62.5)**
Yes	18 (24.7)	6 (15.4)	8 (12.1)	**4 (50.0)**	**4 (17.4)**	**3 (37.5)**
Had transactional sex						*****
No	64 (87.7)	34 (87.2)	64 (97.0)	**5 (62.5)**	**22 (95.7)**	**6 (75.0)**
Yes	9 (12.3)	5 (12.8)	2 (3.0)	**3 (37.5)**	**1 (4.3)**	**2 (25.0)**
Has a boyfriend						
No	70 (95.9)	37 (94.9)	66 (98.5)	8 (100)	21 (91.3)	7 (87.5)
Yes	3 (4.1)	2 (5.1)	1 (1.5)	0 (0)	2 (8.7)	1 (12.5)
Vaginal Community State Type (CST)			*******			
CST-I (*L. crispatus* dominated)	**29 (42.6)**	**27 (79.4)**	**45 (70.3)**	2 (28.6)	11 (50.0)	1 (14.3)
CST-III (*L. iners* dominated)	**32 (47.1)**	**6 (17.6)**	**15 (23.4)**	4 (57.1)	11 (50.0)	4 (57.1)
CST-IV (mixed)	**4 (10.3)**	**1 (2.9)**	**4 (6.3)**	1 (14.3)	0 (0)	2 (28.6)
*Characteristics at Follow-Up*						
Median age			*******			
Below 18.8 years	**33 (45.2)**	**31 (79.5)**	**31 (46.3)**	4 (50.0)	15 (65.2)	5 (62.5)
18.8 years or older	**40 (54.8)**	**8 (20.5)**	**36 (53.7)**	4 (50.0)	8 (34.8)	3 (37.5)
Socioeconomic score						
Above median	35 (48.6)	23 (60.5)	44 (65.7)	5 (62.5)	16 (69.6)	5 (71.4)
Below median	37 (51.4)	15 (39.5)	23 (34.3)	3 (37.5)	7 (30.4)	2 (28.6)
Sexually active						
No	29 (40.3)	18 (47.4)	28 (42.4)	2 (28.6)	7 (30.4)	3 (42.9)
Yes	43 (59.7)	20 (52.6)	38 (57.8)	5 (71.4)	16 (69.6)	4 (57.1)
Experienced coerced sex						
No	58 (81.7)	34 (89.5)	59 (88.1)	7 (87.5)	16 (69.6)	5 (71.4)
Yes	13 (18.3)	4 (10.5)	8 (11.9)	1 (12.5)	7 (30.4)	2 (28.6)
Had transactional sex						
No	63 (87.5)	35 (92.1)	63 (94.0)	7 (87.5)	18 (78.3)	6 (85.7)
Yes	9 (12.5)	3 (7.9)	4 (6.0)	1 (12.5)	5 (21.7)	1 (14.3)
Has a boyfriend						
No	55 (76.4)	35 (92.1)	52 (77.6)	4 (50.0)	16 (69.6)	4 (57.1)
Yes	17 (23.6)	3 (7.9)	15 (22.4)	4 (50.0)	7 (30.4)	3 (42.8)
Vaginal CST at follow-up			*******			*******
CST-I (*L. crispatus* dominated)	**2 (2.8)**	**36 (92.3)**	**51 (78.5)**	**0 (0)**	**7 (30.4)**	**0 (0)**
CST-III (*L. iners* dominated)	**50 (69.4)**	**3 (7.7)**	**13 (20.0)**	**1 (12.5)**	**14 (60.9)**	**7 (87.5)**
CST-IV (mixed)	**20 (27.8)**	**0 (0)**	**1 (1.5)**	**7 (87.5)**	**2 (8.7)**	**1 (12.5)**
Etiology (restricted to single infections)						
*C. trachomatis* (n = 28)				2 (28.6)	13 (61.9)	6 (85.7)
*N. gonorrhoeae* (n = 4)				0 (0)	2 (9.5)	1 (14.3)
*T. vaginalis* (n = 12)				5 (71.4)	6 (28.6)	0 (0)

Chi-square *p*-value; Fisher’s test applied where cell size < 5. * *p*-value < 0.05; *** *p*-value < 0.001. Comparisons with *p*-value < 0.05 are bolded.

## Data Availability

This study was conducted with approval from the Kenya Medical Research Institute (KEMRI) Scientific and Ethics Review Unit (SERU), which requires that data be released from any KEMRI-based Kenyan studies (including de-identified data) only after their written approval for additional analyses. In accordance, data for this study will be available upon request, after obtaining written approval for the proposed analysis from the KEMRI SERU. Their application forms and guidelines can be accessed at https://www.kemri.org/seru-overview (last accessed 3 August 2023). To request these data, please contact the KEMRI SERU at seru@kemri.org.

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
