# Peer review of "Vaginal Microbial Network Analysis Reveals Novel Taxa Relationships among Adolescent and Young Women with Incident Sexually Transmitted Infection Compared with Those Remaining Persistently Negative over a 30-Month Period"

_microorganisms, 2023, doi:10.3390/microorganisms11082035_

Round 1
Reviewer 1 Report
Overall, the study is well-conducted, and I don't have any major issues. However, I do have some inquiries for the authors:
Ethical implications and risks: It is important for the authors to address the potential ethical implications and risks associated with publishing sensitive information about young females' sexual activity in the region. They should acknowledge the potential harm to privacy and reputation, discuss mitigation strategies, and reflect on responsible dissemination. Considering these factors is crucial to protect participants and minimize potential negative consequences in the community.
Categorization of sexual behaviors: To enhance the study's focus on the relationship between behavior and the vaginal microbiome, I would expect a different categorization. Instead of "Ever Sexually active," "Had transactional sex," "Experienced coerced sex," and "Has a boyfriend," it would be more insightful to consider the following classifications: "No sexual activity," "Monogamous sexual activity" (indicating exclusive sexual involvement with one partner), and "Polygamous sexual activity" (representing engagement in sexual activity with multiple partners). These suggested categories would provide more meaningful insights into the impact of different sexual behaviors on the vaginal microbiome.
Attention to L. jensenii in the incident STI group: The identification of L. jensenii among the highest centralities in the microbiome of the incident STI group is notable. This observation is significant because L. jensenii is known to play a protective role and is generally considered a healthy bacterium. It would be helpful if the authors expanded a little on the interpretation of this finding in the discussion.
Figure 2C visibility: The pie charts in Figure 2C appear small and could be improved to be more visible. Enhancing the size or clarity of the pie charts would enhance the readability and understanding of the figure.
Additionally, the quality of the figures 2, 3, and 4 (colors) could be better or of better resolution and higher quality.
Definition of abbreviation: The abbreviation BVAB1 is not defined in the text.
No comments regarding the English quality
Author Response
Reviewer 1
Comments and Suggestions for Authors
Overall, the study is well-conducted, and I don't have any major issues. However, I do have some inquiries for the authors:
- Thank you for your supportive and helpful review.
Ethical implications and risks: It is important for the authors to address the potential ethical implications and risks associated with publishing sensitive information about young females' sexual activity in the region. They should acknowledge the potential harm to privacy and reputation, discuss mitigation strategies, and reflect on responsible dissemination. Considering these factors is crucial to protect participants and minimize potential negative consequences in the community.
- The reviewer raises an important point, and this is also why we do not report the specific areas or schools in which the research took place. We have added to the Limitations points regarding stigma and likely underreporting, and that during study procedures, extensive efforts were made to ensure confidentiality and privacy (as published in our prior paper [12]).
Categorization of sexual behaviors: To enhance the study's focus on the relationship between behavior and the vaginal microbiome, I would expect a different categorization. Instead of "Ever Sexually active," "Had transactional sex," "Experienced coerced sex," and "Has a boyfriend," it would be more insightful to consider the following classifications: "No sexual activity," "Monogamous sexual activity" (indicating exclusive sexual involvement with one partner), and "Polygamous sexual activity" (representing engagement in sexual activity with multiple partners). These suggested categories would provide more meaningful insights into the impact of different sexual behaviors on the vaginal microbiome.
- We agree – the variable “sexually active” is very broad. Being sexually active was assessed with a series of questions (willingly had sex, have sex with someone considered to be a boyfriend/partner/husband, coerced or forced to have sex, or sex under various transactional settings – in exchange for money, goods, food, favors, etc.). For sensitivity to participants, the number of sex partners was only asked for those who disclosed directly to being sexually active (i.e., not assessed for each of the transactional sex settings). Therefore, there are many participants (52%) who reported having sexual activity, but for whom the number of sex partners is not assessed/missing. Further, of those who reported being sexually active, there was substantial underreporting of number of sex partners. For example, 7 STI infections were among 28 participants reporting “zero” sex partners in the past 6 months. We have previously reported on the underreporting of sexual activity in [12], and believe this ties in with the importance of the prior Reviewer 1 comment, which we have now drawn attention to in the Limitations.
- Of note, adolescents with boyfriends may or may not be sexually active. While most (97%) participants with a boyfriend also reported being sexually active, just 42% of sexually active participants had a boyfriend.
Attention to L. jensenii in the incident STI group: The identification of L. jensenii among the highest centralities in the microbiome of the incident STI group is notable. This observation is significant because L. jensenii is known to play a protective role and is generally considered a healthy bacterium. It would be helpful if the authors expanded a little on the interpretation of this finding in the discussion.
- The reviewer raises an important point. jensenii generally has a protective role and we have added information and citations on this.
- As relates to the centrality, it is unclear whether the role of jensenii is present prior to infection or as a result of infection. L. jensenii did not have high centrality in those who remained persistently negative for BV and STI, so one might hypothesize its centrality changes in response to infection. However, longitudinal network studies will be necessary to disentangle this and we have added this point to the Discussion.
Figure 2C visibility: The pie charts in Figure 2C appear small and could be improved to be more visible. Enhancing the size or clarity of the pie charts would enhance the readability and understanding of the figure.
- We have increased the size of the fonts and pie charts in Figure 2.
Additionally, the quality of the figures 2, 3, and 4 (colors) could be better or of better resolution and higher quality.
- The high resolution images were submitted to the journal as separate files and it is likely that the embedded image files are not what will be published. Nevertheless, we have attempted to make the colors, figures, and fonts larger/more visible.
Definition of abbreviation: The abbreviation BVAB1 is not defined in the text.
- Apologies for this oversight. We have added the definition in the text and in Table 3.
Reviewer 2 Report
The authors characterized the microbiome of different groups of bacteria relate to incident STIs, especially among adolescents and try to apply the microbial network analysis to identify key taxa. On the whole, I think the paper could merit publication in Microorganisms, but after major revision.
1. There is limited description of species composition and abundance of the vaginal microbiome from each of sampling groups. The microbial network analysis did not indicate the interspecies metabolic cross-talks in the context of diversity of microbial species and metabolite conversion and consumption of one species to another within the microenvironment. At the species level, there is no complete functional analysis of microbes. As a result, this data is preliminary and should be investigated further in terms of niche.
2. p 8 the subtitle below the legend of Figure 2 is too long.
3. p 10 what’s the meaning of “Centralities (Table 3).”?
Author Response
Reviewer 2
Comments and Suggestions for Authors
The authors characterized the microbiome of different groups of bacteria relate to incident STIs, especially among adolescents and try to apply the microbial network analysis to identify key taxa. On the whole, I think the paper could merit publication in Microorganisms, but after major revision.
- There is limited description of species composition and abundance of the vaginal microbiome from each of sampling groups. The microbial network analysis did not indicate the interspecies metabolic cross-talks in the context of diversity of microbial species and metabolite conversion and consumption of one species to another within the microenvironment. At the species level, there is no complete functional analysis of microbes. As a result, this data is preliminary and should be investigated further in terms of niche.
- We thank the Reviewer for their helpful comments and suggestions.
- Regarding description of species composition and abundance of the vaginal microbiome from each of the sampling groups: In addition to Figure 1 showing the relative abundance of the 10 taxa with the highest mean relative abundance for the entire sample, Figure 2 shows the distribution of community state type and relative abundance of 10 highest mean abundance taxa by sampling group. Lastly, Supplemental Table 3 shows the presence/absence and mean RA for all 54 taxa by each sampling group. We apologize if Supplemental Table 3 was not within the Reviewer’s materials received.
- Reviewer is correct that we did not include the interspecies metabolic cross-talks in the context of diversity, or metabolite conversion and consumption, nor any functional analysis. As we note in the Discussion, such studies of bacterial function integrated with network composition and sociodemographic and behavioral data are urgently needed. Such substantial undertaking is out of scope for the current analysis, but we are considering ways to incorporate this in our study going forward.
- p 8 the subtitle below the legend of Figure 2 is too long.
- We defer to the journal on acceptable length for the legend.
- p 10 what’s the meaning of “Centralities (Table 3).”?
- We apologize for mislabeling. The centrality values are presented in columns 2 and 3, and the first column is the taxa. We have revised the headings to correct this. The definitions of the centralities are presented in the Methods.